# Impact of Invasive Pulmonary Aspergillosis in Critically Ill Surgical Patients with or without Solid Organ Transplantation

**DOI:** 10.3390/jcm12093282

**Published:** 2023-05-04

**Authors:** Simon Dubler, Michael Etringer, Markus A. Weigand, Thorsten Brenner, Stefan Zimmermann, Paul Schnitzler, Bettina Budeus, Fabian Rengier, Paulina Kalinowska, Yuan Lih Hoo, Christoph Lichtenstern

**Affiliations:** 1Department of Anaesthesiology, Heidelberg University Hospital, D-69120 Heidelberg, Germany; 2Department of Anesthesiology and Intensive Care Medicine, University Hospital Essen, University Duisburg-Essen, D-45147 Essen, Germany; 3Translational Lung Research Center Heidelberg (TLRC), Member of the German Center for Lung Research (DZL), University of Heidelberg, D-69120 Heidelberg, Germany; 4Department of Infectious Diseases, Medical Microbiology and Hygiene, Division Bacteriology, Heidelberg University Hospital, D-69120 Heidelberg, Germany; 5Department of Infectious Diseases, Virology, Heidelberg University Hospital, D-69120 Heidelberg, Germany; 6Institute of Cell Biology (Cancer Research), University of Duisburg-Essen, D-45147 Essen, Germany; 7Clinic for Diagnostic and Interventional Radiology, Heidelberg University Hospital, D-69120 Heidelberg, Germany

**Keywords:** aspergillosis, solid organ transplantation, critical care, intensive care unit, immunosuppression

## Abstract

Background: Critically ill patients, especially those who have undergone solid organ transplantation (SOT), are at risk of invasive pulmonary aspergillosis (IPA). The outcome relevance of adequately treated putative IPA (pIPA) is a matter of debate. The aim of this study is to assess the outcome relevance of pIPA in a cohort of critically ill patients with and without SOT. Methods: Data from 121 surgical critically ill patients with pIPA (*n* = 30) or non-pIPA (*n* = 91) were included. Cox regression analysis was used to identify risk factors for mortality and unfavourable outcomes after 28 and 90 days. Results: Mortality rates at 28 days were similar across the whole cohort of patients (pIPA: 31% vs. non-pIPA: 27%) and did not differ in the subgroup of patients after SOT (pIPA: 17% vs. non-pIPA: 22%). A higher Sequential Organ Failure Assessment (SOFA) score and evidence of bacteraemia were identified as risk factors for mortality and unfavourable outcome, whereas pIPA itself was not identified as an independent predictor for poor outcomes. Conclusions: Adequately treated pIPA did not increase the risk of death or an unfavourable outcome in this mixed cohort of critically ill patients with or without SOT, whereas higher disease severity and bacteraemia negatively affected the outcome.

## 1. Introduction

The number of invasive fungal infections (IFI) has increased in recent years. After *Candida* spp., *Aspergillus* spp. are the second most commonly isolated pathogens, accounting for 0.3–19% of all IFI [1]. This is mainly due to the increasing proportion of multi-morbid, elderly and immunocompromised patients [2]. Solid organ transplant (SOT) recipients have one of the highest risks of developing invasive aspergillosis (IA) [3]. The lungs are the most affected organ in these patients, and an infection with *Aspergillus* spp. is called invasive pulmonary aspergillosis (IPA) [4]. This disease shows an incidence rate of up to 59%, and the mortality rate exceeds >50% in patients after SOT [2,5,6].

In recent years, IPA has increased in other non-immunocompromised patient populations (patients with chronic obstructive pulmonary disease (COPD), liver diseases, congestive heart failure, diabetes, kidney failure and weight loss [7]), namely critically ill patients in the intensive care unit (ICU) [8], and especially those with severe respiratory viral infections caused by influenza virus or severe acute respiratory syndrome coronavirus 2 (SARS-CoV-2) [9,10]. The incidence of IPA in these non-immunocompromised, critically ill patients is not precisely known because of diagnostic difficulties. Indeed, the differentiation of an invasive infection from a pure colonisation often fails due to the unfeasibility of a histopathological diagnosis. Additionally, non-neutropenic, non-immunocompromised patients regularly lack classical host factors (e.g., allogenic stem cell recipient, inherited severe immunodeficiency or a history of neutropenia) and radiological features (e.g., cavity or air-crescent sign) [8]. To complicate matters even more, non-immunocompromised patients show a decreased validity for non-culture-based methods such as galactomannan (GM) [11]. An Italian multicentre observational study examined the presence of IFI in 18 ICUs over a period of 18 months. A total of 105 episodes of IFI occurred in 5561 patients. Only 12 patients were diagnosed with invasive pulmonary aspergillosis, but mortality was as high as 60% [12].

Given the above-mentioned situation, the aims of the present study are to distinguish between putative invasive IPA (pIPA) and a pure colonisation (non-pIPA), and to assess the relevance of pIPA on the outcome in a cohort of critically ill patients with and without SOT.

## 2. Methods

### 2.1. Study Design and Patient Population

This monocentric, retrospective study was conducted in a surgical ICU at the University Hospital Heidelberg (Heidelberg, Germany) between March 2015 and December 2019. The University Hospital Heidelberg is a tertiary medical and referral centre for SOT (mostly kidney and liver transplantations). This study was approved by the local ethics committee of the Medical Faculty of the University of Heidelberg (S-191/2018) and was registered at the German clinical trials register (DRKS-ID: DRKS00024735).

Adult patients (>18 years) in the surgical ICU with GM ≥ 1.0 optical density index (ODI) according to the manufacturer’s instructions (Platelia™ *Aspergillus* ELISA, Bio-Rad, Marnes-la-Coquette, France) from bronchoalveolar lavage fluid (BALF) and with at least one computed tomography (CT) scan of the chest or chest X-ray were included in the study. GM in patients’ sera was measured only fragmentarily and was therefore not included in the presented analysis. Indications of bronchoalveolar lavage (BAL) were a decrease in pulmonary status (oxygenation or ventilation failure) or suspected sepsis. The investigation followed the local standardised sepsis protocols [13] consisting of empirical antibacterial therapy, sterile drawing of blood cultures and BAL/CT or chest x-ray, if indicated. Data consisting of focus control (abdominal surgery, change of indwelling catheters, etc.) were extracted. Given the retrospective nature of the data analysis, it was not possible to identify certain septic focus/control measurements in every case.

Patients with orthotropic liver transplantation (OLT) were treated based on local standard operating procedures (SOP) for OLT [14]. This includes a standardised immunosuppressive regimen consisting of corticosteroids and calcineurin inhibitors (tacrolimus or cyclosporine). After SOT, each patient received fluconazole for antifungal prophylaxis. High-risk patients (MELD (Model for Endstage Liver Disease) >30; Re-liver transplantation and patients after high-urgency (HU) transplantation) received caspofungin for antifungal prophylaxis. Valganciclovir was used for cytomegalovirus (CMV) prophylaxis. Patients receiving haemodialysis due to chronic or new kidney insufficiency received ganciclovir.

### 2.2. Patient Data

The following data were collected from the electronic medical records (ISH^®^, SAP, Walldorf, Germany): demographics, past medical history (including chronic obstructive pulmonary disease (COPD), diabetes mellitus, coronary artery disease, heart insufficiency, alcohol abuse, liver cirrhosis, history of solid tumour disease, history of cytotoxic substances (chemotherapy), SOT, history of stroke as well as complications during hospitalisation (including candidaemia, bacteraemia, dialysis, delirium, corticosteroid therapy and time of mechanical ventilation)). To assess the severity of illness, the American Society of Anesthesiologists (ASA) classification and the Sequential Organ Failure Assessment (SOFA) score were collected at different time points.

### 2.3. Outcomes

The primary endpoint was 28-day mortality (from any cause) after ICU admission. Secondary endpoints were an unfavourable outcome, including death or persistent ICU stay, at 28 and 90 days after ICU admission.

### 2.4. Definitions

#### 2.4.1. IPA

IPA was defined according to the AspICU criteria (Appendix A) [15]. In addition to the AspICU criteria, a GM ≥ 1.0 in a BALF specimen also served as an entry criterion for IPA diagnosis [16]. Patients fulfilling all four criteria (clinical data + radiological findings + host factors + mycological findings) were termed ‘putative IPA’ (pIPA). Patients not fulfilling these criteria (more than 1 criterion missing) were termed ‘non-pIPA’ (respiratory tract colonisation only) (Appendix A).

#### 2.4.2. Second-Line and Rescue Therapy

The decision to switch to another fungal agent or to add a new antifungal agent was driven by clinical evidence (e.g., decrease in respiratory status), mycological evidence (e.g., direct (cultured) or indirect (GM) evidence of *Aspergillus* spp.) in BALF, or radiological evidence (e.g., ongoing radiological evidence of IPA on chest CT) by the consultant in charge.

#### 2.4.3. Corticosteroid Therapy

Corticosteroid therapy comprised the use of prednisolone >20 mg/day or another corticosteroid at an equivalent dosage during or before the hospital stay prior to the first GM detection in BALF.

### 2.5. Microbiology

GM testing was performed with the Platelia™ Aspergillus ELISA (Bio-Rad). Microbial growth in blood culture bottles (BACTEC FX^®^ Aerobic/F (Ref. 442023), Lytic/10 Anaerobic/F (Ref. 442021), BD Diagnostics, Sparks, NV, USA) was detected using the BACTEC FX^®^ automated blood culture system (BD Diagnostics) and subsequently confirmed via Gram staining. Positive cultures were worked up according to approved in-hospital standard techniques. Briefly, Columbia blood agar, chocolate agar and McConkey agar was used for plating of positive blood culture bottles; for Lytic/10 Anaerobic/F bottles an additional Schaedler/KV biplate (PB5204E, Thermofisher, Wesel, Germany) was inoculated. If yeast or filamentous fungi were seen in the Gram stain, an additional CHROMagar™ Candida (254106, Becton Dickinson, Sparks, NY, USA) or Sabouraud Chloramphenicol™ agar (254091, BD) was inoculated and incubated for 24–72 h, respectively.

*Aspergillus* spp. isolates were grown from respiratory specimens. Isolates were investigated at the Department of Medical Microbiology and Hygiene, bacteriology division of University Hospital Heidelberg, Heidelberg, Germany. Briefly, all respiratory specimens were inoculated on Sabouraud Chloramphenicol™agar (254091, BD) and incubated at 30 °C degrees for 7 days with daily plate inspections. Only positive cultures with *Aspergillus* spp. were included in the study. For yeast identification, MALDI-TOF/MS identification was performed, as described previously [17]. For filamentous fungi, a PCR for the *its* region was performed followed by classical sequencing, as described previously [18,19].

### 2.6. BAL

BAL for invasively ventilated or awake patients was performed according to a local standardised protocol (‘wedging’ both sides of the lung with 20 mL of sterile sodium chloride (NaCl) 0.9% followed by re-aspiration; 10 mL of each side was used for further diagnostics). Non-intubated patients received BAL while awake.

### 2.7. Statistics

The data were stored in Excel (Microsoft^®^, Redmond, WA, USA) and then analysed in R (R Core Team (2022) [20]) using the packages survminer (Kassambara, A. et al. [21]), R package version 0.4.9), and survival (Therneau, T. (2021) [22] R package version 3.2–13), for survival analysis, and gtsummary [23] to display the data in tables. Continuous data are presented as the mean and standard deviation. Categorical variables are displayed as absolute and relative frequencies. The Mann–Whitney U-test or the chi-square test was used to calculate potential differences between the groups. Kaplan–Meier curves present survival information. Cox proportional hazards regression model with adjustment for potential confounders was used to identify risk factors for mortality and unfavourable outcome (hazard ratio (HR)). Two-sided *p* < 0.05 was considered statistically significant for all analyses.

## 3. Results

### 3.1. Patient Demographics and Baseline Characteristics

During the 5-year observation period, a total of 121 patients could be identified for the final analysis. According to the aforementioned criteria, 30/121 (24.8%) and 91/121 (75.2%) patients were categorised as pIPA and non-pIPA (pure colonisation), respectively. The baseline characteristics of all included patients are displayed in detail in Table 1. Patients with pIPA displayed more underlying liver disease with liver cirrhosis than did non-pIPA patients (50% vs. 23%, *p* = 0.005), and/or a history of SOT (63% vs. 25%, *p* < 0.001).

Corticosteroid use was higher among patients with pIPA (97% vs. 52%, *p* ≤ 0.001). There were also differences in terms of the American Society of Anesthesiologists (ASA) status (*p* = 0.03) and Sequential Organ Failure Assessment (SOFA) scores at ICU admission between pIPA- and non-pIPA patients (11 vs. 8, *p* = 0.04). The length of ICU stay and the duration of mechanical ventilation tended to be prolonged in pIPA patients compared to non-pIPA patients (46 vs. 30 days (*p* = 0.065) and 453 vs. 270 h (*p* = 0.12), respectively).

Patients with pIPA presented with higher SOFA scores at first GM detection in BALF (GM-SOFA) (13 vs. 10, *p* = 0.002). The first GM value in BALF and the maximum GM value in BALF during ICU stay was also significantly higher (4.5 vs. 1.18, *p* ≤ 0.001 and 5.84 vs. 1.21, *p* ≤ 0.001, respectively) in patients with pIPA.

In SOT recipients, the GM-SOFA scores did not differ significantly between patients with pIPA and non-pIPA patients (*p* = 0.3). GM values in BALF (first or highest) were significantly different between pIPA and non-pIPA patients for SOT recipients (first GM in BALF: 4.60 vs. 1.66, *p* = 0.005 and highest GM in BALF: 6.36 vs. 1.66, *p* ≤ 0.001, respectively).

There was also a difference between patients with pIPA and non-pIPA patients in terms of positive *Aspergillus* spp. cultures in BALF (47% vs. 23%, *p* = 0.01). Most frequently, *A. fumigatus* (*n* = 29) alone could be cultured. Other *Aspergillus* spp. or mixed cultures were only observed very rarely: *A. fumigatus*/*A. flavus* (*n* = 2), *A. flavus* (*n* = 2), *A. fumigatus*/*A. terreus* (*n* = 1) and *A. fumigatus*/*A. delacroxii*/*A. nidulans* (*n* = 1).

There was no difference in the use of antifungal prophylaxis between pIPA- and non-pIPA patients (23% vs. 20%). First-line, second-line and rescue antifungal therapies were more frequently administered to patients with pIPA than to non-PIPA patients (90% vs. 60%, *p* = 0.003; 50% vs. 15%, *p* ≤ 0.001; and 20% vs. 4.4%, *p* = 0.02). Table 2 shows the microbiological test results and corresponding antifungal treatments in all patients (see Appendix A for patients with and without SOT). For first-line therapy, voriconazole was the most frequently used antifungal agent, followed by liposomal amphotericin-B. Azols (voriconazole as well as isavuconazole) and liposomal amphotericin-B, which were used equally for second-line therapy. Rescue therapy strategies consisted of a monotherapy (posaconazole or micafungin) or combination therapies. Appendix A shows detailed information about the antifungal treatment strategies.

### 3.2. Outcome

The primary outcome, i.e., mortality within 28 days after ICU admission, did not differ significantly between patients with pIPA and non-pIPA patients (pIPA: 31% (9/30) vs. 27% (24/91), *p* = 0.7; Appendix A). This also holds true for the SOT subgroup (pIPA: 17% (3/19) vs. 22% (5/23), *p* = 0.9; Appendix A) and the non-SOT subgroup (pIPA 55% (6/11) vs. 29% (19/68), *p* = 0.2; Appendix A). Moreover, there was no significant difference in mortality after 28 days between SOT recipients (26% total, 20% died) and patients who did not receive SOT (74% total, 34% died) (*p* = 0.09). However, when comparing the 28-day mortality of patients with pIPA, there was a significant difference between both groups (*p* = 0.04, 63% received SOT; 16% of those recipients died, whereas more than 54% of the patients who did not receive SOT died). In non-pIPA patients, there was no significant difference between the two groups (*p* = 0.5, 25% received SOT, 22% died; 28% of patients who did not receive SOT died). Table 3 shows a more detailed presentation of the other outcome variables for all the patients including the SOT and non-SOT subgroups.

To identify risk factors for mortality and unfavourable outcomes with adjustment for potential confounders, the Cox proportional regression model was used. Higher SOFA scores (increase of 1 point) at first GM detection in BALF were associated with a relative increase in 28-day mortality risk of 1.25 (95% confidence interval (CI) 1.11–1.40) in all patients. Suffering from bacteraemia during the ICU stay increased the mortality risk by 2.94 (95% CI 1.40–6.18) in all the patients included (Table 4). In the non-SOT subgroup, higher SOFA scores at first GM detection in BALF increased the risk of 28-day mortality by 1.33 (95% CI 1.14–1.56) (Appendix A). By contrast, there was no association between higher SOFA scores and an increased 28-day mortality in the SOT subgroup (*p* = 0.206), whereas bacteraemia was again linked to an increased 28-day mortality rate, specifically by 6.14 (95% CI 1.32–28.73) (Appendix A).

Concerning secondary outcomes, higher SOFA scores (HR 1.12, 95% CI 1.06–1.19) and the evidence of bacteraemia (HR 2.18, 95% CI 1.33–3.59) were associated with an increased risk of an unfavourable outcome within 28 days for all the included patients (Appendix A). In the non-SOT subgroup, bacteraemia (HR 2.70, 95% CI 1.41–5.16) and higher SOFA scores (HR 1.18, 95% CI 1.08–1.29) were associated with an increased risk of an unfavourable outcome within 28 days (Appendix A). The risk factors for death or unfavourable outcomes within 90 days after ICU admission are shown in Appendix A. Suspected infectious foci in patients with bacteraemia are displayed in Appendix A.

Even after adjusting for confounders, pIPA itself was not an independent risk factor for mortality or an unfavourable outcome after 28 or 90 days when considering all the patients, nor was it such a factor in the SOT and non-SOT subgroups (Table 4 and Appendix A).

## 4. Discussion

In this retrospective multivariate analysis, we describe the prevalence and outcome of pIPA versus non-pIPA in surgical critically ill patients with and without SOT. Since a pharmacologically induced state of immunosuppression (e.g., after SOT) is a well-known risk factor for IPA, SOT recipients in the presented analysis had pIPA more frequently compared with patients who did not undergo SOT. However, under the prerequisite of early diagnosis and timely treatment, pIPA could not be identified as an independent risk factor for mortality or an unfavourable outcome within 28 or 90 days in the SOT and non-SOT subgroups, even after adjusting for various confounders.

IPA is uncommon in patients without underlying co-morbidities (e.g., diabetes mellitus, liver cirrhosis, COPD, alcohol abuse, dialysis, non-haematological malignancy), although these factors do not belong to the documented classic risk factors [15]. Taccone et al. [24] found that only 5% of 297 ICU patients with IPA did not have any co-morbidities. In our cohort, conditions such as diabetes mellitus, liver cirrhosis, COPD, non-haematological malignancies, and dialysis were common in patients with pIPA. In the AspICU cohort of non-immunocompromised critically ill patients [8], those with proven or putative IPA showed higher SOFA scores (11 and 9 points) compared with patients with a colonisation only (5 points). In another study in patients with influenza [9], higher APACHE II scores, as a marker of disease severity, were independently associated with IPA. Accordingly, GM-SOFA scores in the whole cohort of patients in the present study were also significantly higher in the pIPA group than in the non-pIPA group, but GM-SOFA scores did not differ significantly between the SOT and non-SOT subgroups. In the literature, SOFA scores of 10 points correlate with a mortality rate of 40%, and those higher than 11 points correlate with mortality rates greater than 80% [25]. Accordingly, GM-SOFA was an independent risk factor for 28-day mortality after the multivariate analysis of all participating patients, including the non-SOT subgroup. This might reflect the high disease burden in this mixed cohort of critically ill patients. Bacteraemia during the ICU stay was another risk factor for mortality and an unfavourable outcome after 28 and 90 days in all the patients. Of the 27 patients with bacteraemia in our cohort, almost half of them (44% (12/27)) revealed an abdominal focus. This is not surprising, since the cohort consisted of critically ill surgical patients at risk from bacteraemia [26] due to postoperative surgical complications such as wound infections, anastomotic leakage, intra-abdominal abscesses or bowel ischemia with lethal complications [27]. This might have contributed to the high burden of bacteraemia in our cohort.

Since our cohort consisted of ICU patients with and without SOT, the EORTC/MSG criteria [28] seem to be far from perfect for IPA diagnosis for several reasons, including the following: (1) Lung biopsies are not feasible in critically ill patients in the ICU due to bleeding or respiratory complications. (2) Non-immunocompromised ICU patients most often lack the ‘host’ criteria according to the consensus definitions, but are still at risk due to underlying co-morbidities, acute organ failure, sepsis, or septic shock.

Positive *Aspergillus* spp. cultures from BALF differed in patients with pIPA and non-pIPA patients, whereas 28-day outcomes were equal in both groups. Of the SOT recipients, only 47% of patients with pIPA showed positive *Aspergillus* spp. cultures (vs. 23% in non-pIPA patients). Considering the non-SOT subgroup, 45% of patients with pIPA showed positive *Aspergillus* spp. cultures (vs. 25% in non-pIPA patients). This reflects another diagnostic weakness, because culture-based diagnostics have represented a cornerstone of IPA diagnosis. However, studies have shown a sensitivity of ≤65% for cultural positivity in the diagnosis of IPA [29], given that most patients cannot be diagnosed by culture-based diagnostics. Therefore, alternative diagnostic tools (e.g., GM) are necessary to overcome this diagnostic dilemma [30]. The original AspICU diagnostic algorithm by Blot et al. [15] was used in our study. However, to apply this regimen to a broader spectrum of ICU patients (and to overcome the diagnostic dilemma), we also included patients with a GM ≥ 1.0 (ODI) from BALF as proposed by Schroeder et al. [16]. GM is a cell wall component (polysaccharide) of *Aspergillus* spp. that is released by growing hyphae or germinating conidia and can then be measured in blood or BALF. Blood GM (cut-off > 0.5 ODI) was intended to serve as a screening tool for IA in high-risk patients (haematological malignancies and SOT recipients). However, due to its low sensitivity of around 30% for IPA diagnosis in non-neutropenic patients in the ICU, it is not as useful for analytic purposes. The performance characteristics of GM in BALF seem to be more beneficial, but evidence-based and well-accepted limit values are still lacking.

Due to missing recommendations in the 2008 EORTC/MSG guidelines [28], cut-off values from the manufacturer (Platelia™ *Aspergillus*, Bio-Rad) were used (ODI of 0.5) for GM in BALF. The Platelia assay was initially only approved for serum; BALF was added and validated later. Moreover, there are limitations regarding GM due to occurrences of false positivity from co-medications; underlying diseases; or aspiration [29,31]. In light of increasing evidence over the years, the updated EORTC/MSG consensus definitions in 2019 included a cut-off ODI of 1.0 in BALF [32]. A meta-analysis showed better specificity with ODI 1.0 (0.94–0.95) compared with ODI 0.5 (0.89–0.92), with only a minor decrease in sensitivity (0.82–0.87 vs. 0.75–0.86) [33]. Among the other major updates to the ‘probable’ IPA definition in 2019, the following was included: ‘Radiographic features’ are less specific compared with the 2008 guidelines and include no wedge-shaped, segmental or lobar consolidations and a reverse halo sign.

We therefore decided to include patients with a GM ODI of ≥1.0 in BALF. The association between GM (BALF) and mortality rates has recently been investigated in patients with COVID-19-associated pulmonary aspergillosis (CAPA). Bartoletti et al. [34] found a correlation between the magnitude of GM in BALF and 30-day mortality. The odds of death within 30 days of ICU admission increased 1.41-fold for each point increase in the initial GM in BALF. This is in accordance with the data presented in this study: A higher GM value in BALF during ICU stay was associated with a worse outcome after 28 days. GM values in BALF tended to be higher in SOT recipients compared to the non-SOT subgroup. Higher GM values might be an indicator of a higher fungal burden. By contrast, an inability to reduce GM values over time might reflect a patient’s compromised immune status, which can also lead to a change in or prolonged antifungal therapy. In line with the latest IDSA guidelines [35], voriconazole was used most frequently as a first-line antifungal agent, followed by liposomal amphotericin-B. Second-line therapy included azols, echinocandins or liposomal amphotericin B. By contrast, rescue therapy was quite heterogenous. Recent IDSA guidelines [35] give combination treatment with voriconazole and an echinocandin only as a weak recommendation in selected patients (level 2c recommendation). Liposomal amphotericin B or posaconazole might be considered an option for salvage therapy (level 2a recommendation).

It is difficult to compare these results to the current literature for several reasons. First, there is no clear definition of ‘salvage therapy’ or ‘rescue therapy’ in the literature. Second, there is no accepted definition of ‘treatment failure’. Third, to our knowledge, there are no existing large, randomised trials including both immunocompromised and non-immunocompromised patients comparing single versus combined antifungal treatment strategies. Studies comparing different treatment regimens in patients with underlying haematological malignancies or stem cell transplantation often use ‘clinical response rates’ as outcome measures. Aside from clinical and bronchoscopy evaluation, this often incorporates ‘complete or partial resolution of radiographic findings’ [36,37]. Critically ill, non-immunocompromised patients with IPA often do not present typical radiological findings consistent with IPA. In an AspICU cohort of 79 ICU patients with proven IPA, 70% of patients did not show radiological findings suggestive of IPA [15]. In a multicentre study conducted in India [38], only a minority of non-immunocompromised patients with invasive mould infections showed suggestive radiological signs (24.1% showed nodules, 6.3% showed the halo sign and 0.8% showed the air-crescent sign).

Considering the aforementioned diagnostic dilemma, we might have missed some ‘real’ patients with pIPA in our SOT and non-SOT subgroups. This suspicion is strengthened by the fact that 74% and 22% of SOT non-pIPA recipients (versus 56% and 13% of the non-SOT non-pIPA patients) received first- and second-line antifungal therapy, respectively. These numbers reflect the diagnostic dilemma that IPA diagnosis clinicians face when treating critically ill patients. However, new diagnostic tools to better differentiate true infections from simple colonisations are on the horizon (e.g., next-generation sequencing (NGS) [39]), and might help to overcome this diagnostic dilemma.

## 5. Conclusions

pIPA is encountered more often in critically ill patients after SOT. Based on our multivariate analysis, pIPA itself does not serve as a risk factor for death or a worse outcome in immunocompetent (non-SOT) or immunocompromised (SOT) critically ill surgical patients. However, this is most likely due to a precise diagnostic and aggressive therapeutic workup rather than an apparent innocuousness of IPA. Thus, great efforts must be devoted to solving this immanent diagnostic dilemma.

## Figures and Tables

**Table 1 jcm-12-03282-t001:** Baseline characteristics of all included patients.

	pIPA (*n* = 30)	Non-pIPA (*n* = 91)	*p*
Demographics			
Age (Years) ^†^	60 (54; 69)	63 (54; 74)	0.7
Female	9 (30)	27 (30)	0.9
Underlying diseases			
COPD	6 (20)	12 (13)	0.4
Coronary artery disease	13 (43)	27 (30)	0.2
Diabetes mellitus	7 (23)	20 (22)	0.9
Heart insufficiency	15 (50)	32 (35)	0.15
Liver cirrhosis	15 (50)	21 (23)	0.005 *
Solid tumour	8 (26.3)	46 (51)	0.2
Chemotherapy	3 (10)	12 (13)	0.8
Solid organ transplantation	19 (63)	23 (25)	<0.001 ***
Liver	13 (43)	18 (20)	0.010 *
Kidney	6 (20)	5 (5.5)	0.03 *
Corticosteroid therapy	29 (97)	47 (52)	<0.001 ***
Because of sepsis	10 (33)	22 (24)	0.3
Bacteraemia	10 (33)	17 (19)	0.1
Candidemia	0 (0)	4 (4.4)	0.6
Dialysis	12 (40)	29 (32)	0.4
Delirium	14 (48)	45 (51)	0.8
Stroke	3 (10)	7 (7.7)	0.7
ASA Status ^‡^			0.03 *
2	0 (0)	11 (12)	
3	17 (57)	57 (63)	
4	12 (40)	23 (25)	
5	1 (3.3)	0 (0)	
SOFA ICU admission ^†^	11 (6; 16)	8 (4; 13)	0.04 *

Abbreviations: COPD (Chronic obstructive pulmonary disease), ASA (American society of Anaesthesiology), SOFA (Sequential organ failure assessment), ICU (Intensive care unit). Data are presented as *n* (%). ^†^ Values are presented as median, (Interquartile range). ^‡^ Values are presented as mean ± standard deviation. * *p* < 0.05; *** *p* < 0.001.

**Table 2 jcm-12-03282-t002:** Microbiological tests and antifungal therapies in all patients.

	pIPA (*n* = 30)	Non-pIPA (*n* = 91)	*p*
SOFA first GM positivity ^†^	13 (10; 15)	10 (8; 13)	0.002 **
BALF-positive GM			
First value of GM (BALF) ^†^	4.50 (2.28; 5.74)	1.18 (0.72; 4.06)	<0.001 ***
Highest value of GM (BALF) ^†^	5.84 (4.50; 7.27)	1.21 (0.72; 4.70)	<0.001 ***
*Aspergillus* spp. culture-positive	14 (47)	21 (23)	0.01 *
Antifungal therapy			
Antifungal prophylaxis	7 (23)	18 (20)	0.7
First line therapy (yes)	27 (90)	55 (60)	0.003 **
Duration first line therapy (d) ^†^	16 (6; 35)	5 (0; 14)	<0.001 ***
Time to first line therapy (d) ^†^	6 (4; 16)	12 (6; 22)	0.2
Second line therapy (yes)	15 (50)	14 (15)	<0.001 ***
Duration second line therapy (d) ^†^	3 (0; 15)	0 (0; 0)	<0.001 ***
Rescue therapy (yes)	6 (20)	4 (4.4)	0.02 *

Abbreviations: SOFA (Sequential organ failure assessment), GM (Galactomannan), BALF (Broncho-alveolar lavage). Data are presented as *n* (%). ^†^ Values are presented as median, (Interquartile range). * *p* < 0.05. ** *p* < 0.01. *** *p* < 0.001.

**Table 3 jcm-12-03282-t003:** Outcome of all included patients.

	pIPA	Non-pIPA	*p*
All included patients (*n*)	30	91	
Time on ICU (d) ^†^	46 (28; 85)	30 (20; 55)	0.065
Mechanical ventilation (h) ^†^	453 (187; 828)	270 (92; 650)	0.12
Death at 28 days	9 (31)	24 (27)	0.7
Non-SOT patients only (*n*)	11	68	
Time on ICU (d) ^†^	31 (22; 46)	28 (18; 45)	0.8
Mechanical ventilation (h) ^†^	332 (305; 828)	346 (86; 648)	0.5
Death at 28 days	6 (55)	19 (29)	0.2
SOT patients only (*n*)	19	23	
Time on ICU (d) ^†^	54 (40; 90)	42 (22; 109)	0.8
Mechanical ventilation (h) ^†^	506 (169; 852)	240 (108; 829)	0.3
Death at 28 days	3 (17)	5 (22)	0.9

Abbreviations: ICU (Intensive care unit), SOT (Solid organ transplantation). Data are presented as *n* (%). ^†^ Values are presented as median, (Interquartile range).

**Table 4 jcm-12-03282-t004:** Risk factors for death within 28 days after ICU admission in all patients.

	Hazard Ratio (95% CI)	*p*
**Univariate Analysis**		
Bacteraemia	3.9 (1.9–7.7)	<0.001 ***
Candidemia	1.7 (0.41–7.2)	0.45
SOFA ICU admission	1 (0.93–1.1)	0.96
GM-SOFA first GM positivity	1.2 (1.1–1.3)	<0.001 ***
First value of GM (BALF)	1.1 (0.97–1.3)	0.12
Highest value of GM (BALF)	1.1 (0.92–1.2)	0.45
Putative IPA	0.83 (0.39–1.8)	0.64
*Aspergillus* spp. culture positivity	0.77 (0.37–1.6)	0.47
Firstline therapy	0.53 (0.23–1.2)	0.13
Time to first line therapy	1 (0.99–1)	0.42
Secondline therapy	0.56 (0.27–1.2)	0.11
Rescue therapy	0.92 (0.28–3)	0.9
Mechanical ventilation	1 (1–1)	0.28
**Multivariate Analysis**		
Bacteraemia	2.94 (1.40–6.18)	0.004 **
Mechanical ventilation	1.00 (1.00–1.00)	0.061
GM-SOFA first GM positivity	1.25 (1.11–1.40)	<0.001 ***
Putative IPA	1.47 (0.63–3.46)	0.376

Abbreviations: SOFA (Sequential organ failure assessment), ICU (Intensive care unit), GM (Galactomannan), BALF (Bronchoalveolar lavage), IPA (invasive pulmonary aspergillosis). ** *p* < 0.01. *** *p* < 0.001.

## Data Availability

Data available on request due to restrictions of privacy or ethical. The data presented in this study are available on request from the corresponding author.

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
