# Peer review of "Impact of Invasive Pulmonary Aspergillosis in Critically Ill Surgical Patients with or without Solid Organ Transplantation"

_jcm, 2023, doi:10.3390/jcm12093282_

Round 1

Reviewer 1 Report

The authors have investigated the effect of pIPA on the clinical outcome in surgical ICU patients with or without SOT. It is a single center retrospective study conducted for the period of March 2015 and December 2019.

Some comments and inquiries:

1.     Line 90. Please define high risk patients and provide relevant reference(s).

2.     Line 127. Were fungal media not used for blood cultures? Using BACTEC system, isolation of Candida glabrata is superior when using fungal media as compared to using aerobic blood culture bottles only. This remains significant for possible missed cases of C. glabrata fungemia if only aerobic culture media are used.

3.     Lines 131-134. Detailed information and references should be provided for the methods used for isolation and identification. Was MALDI-TOF/MS or sequence analysis used for identifiation?

Author Response

Re: Manuscript ID: jcm-2276229

Impact of invasive pulmonary aspergillosis in critically ill surgical patients with or without solid organ transplantation

by

Simon Dubler, Michael Etringer, Markus A. Weigand, Thorsten Brenner, Stefan Zimmermann, Paul Schnitzler, Bettina Budeus, Fabian Rengier, Paulina Kalinowska, Yuan Lih Hoo and Christoph Lichtenstern

Response to the Reviewer`s suggestions                        Essen, 21st April 2023

Dear Editor,

Thank you very much for assessing our manuscript. We are very happy that our manuscript is potentially acceptable for publication in the journal “Journal of Clinical Medicine” after implementation of the latest Reviewer´s suggestions.

We thank both reviewers for their valuable comments, which really helped to improve the quality of our manuscript.

We agree with reviewer 1, that using only aerobic blood culture bottles would result in missing fungemia, especially for C. glabrata. Therefore, we have validated the BD BACTEC Lytic/10 Anaerobic/F bottle for its performance on fungi detection. This blood culture bottle type increased the detection rate of Candida spp. overall. In our lab especially a higher rate of C. glabrata (plus >400%) could be shown (paper submitted).

All issues mentioned by the Reviewer were considered very carefully and we would like to resubmit the revised version of the manuscript. You will find our responses to all issues and a detailed Point-by-Point analysis, explaining how the changes in the revised manuscript have been made.

If you have any further questions or suggestions, please do not hesitate to contact us.

We hope that we have satisfactorily addressed all the issues mentioned by the Reviewer and that the manuscript is now acceptable for publication in the journal “Journal of Clinical Medicine”.

Thank you very much for your helpful suggestions and for your kind support.

With kind regards from Essen

Simon Dubler

------------------------------------------------------------------------------------

Simon Dubler

Department of Anaesthesiology and intensive care medicine

Essen University Hospital

Hufelandstrasse 55

D-45147 Essen

Germany

Phone:                 +49-201 / 723-84053

Fax:                        +49 201 / 723-5949

Mail:                      Simon.Dubler@uk-essen.de

Reviewer #1

Detailed comments

Reviewer´s suggestion:

  1. Line 90. Please define high risk patients and provide relevant reference(s).
  2. Line 127. Were fungal media not used for blood cultures? Using BACTEC system, isolation of Candida glabrata is superior when using fungal media as compared to using aerobic blood culture bottles only. This remains significant for possible missed cases of C. glabrata fungemia if only aerobic culture media are used.
  3. Lines 131-134. Detailed information and references should be provided for the methods used for isolation and identification. Was MALDI-TOF/MS or sequence analysis used for identifiation?

Authors’ statement:

  1. We thank the Reviewer for this important comment and tried to define high risk patients with relevant references.

Revised text section, page 3, line 91 - 92:

High-risk patients [MELD (Model for Endstage Liver Disease) > 30; Re-liver transplantation and patients after high-urgency (HU) transplantation] received caspofungin for anti-fungal prophylaxis.

  1. and 3. We thank the reviewer for these important comments. More details about the methods used were added in the Microbiology section of the methods.

Revised text sections, page 4, line 129 – 148:

GM testing was performed with the Platelia™ Aspergillus ELISA (Bio-Rad). Microbi-al growth in blood culture bottles (BACTEC FX® Aerobic/F (Ref. 442023), Lytic/10 Anaero-bic/F (Ref. 442021), BD Diagnostics, Sparks, United states) was detected by the BACTEC FX® automated blood culture system (BD Diagnostics) and subsequently confirmed by Gram staining. Positive cultures were worked up according to approved in-hospital standard techniques. Briefly, Columbia blood agar, chocolate agar and McConkey agar was used for plating of positive blood culture bottles, for Lytic/10 Anaerobic/F bottles an additional Schaedler/KV biplate (PB5204E, Thermofisher, Germany) was inoculated. If yeast or filamentous fungi were seen in the Gram stain, an additional CHROMagar™ Candida (254106, Becton Dickinson, Sparks, US) or Sabouraud Chloramphenicol™agar (254091, BD) was inoculated and incubated for 24-72 hours respectively.

Aspergillus spp. isolates were grown from respiratory specimens Isolates were inves-tigated at the Department of Medical Microbiology and Hygiene, bacteriology division of University Hospital Heidelberg, Heidelberg, Germany. Briefly, all respiratory specimens were inoculated on Sabouraud Chloramphenicol™agar (254091, BD) and incubated at 30°C degrees for 7 days with daily plate inspections. Only positive cultures with Aspergillus spp. were included in the study.

For yeast identification MALDI-TOF/MS identification was performed, as described previously (17). For filamentous fungi a PCR for the its region was performed followed by classical sequencing as described previously (18, 19).

Reviewer 2 Report

In the manuscript entitled “Impact of invasive pulmonary aspergillosis in critically ill surgical
patients with or without solid organ transplantation”.

The authors have described the invasive pulmonary aspergillosis in critically ill surgical
patients with or without solid organ transplantation. This manuscript is interesting and can be educational. By the way, there are no evidences about results of used direct examination and culture methods!

1.            Lines 28-32: this sentence is unclear, and needs an English revision.

2.            Line 41: “….highest risks of developing invasive aspergillosis (IA).” This sentence doesn’t have any related references. For suggestion, you can use this one too: a. Use of mycological, nested-PCR and real-time PCR methods on BAL fluids for detection of Aspergillus fumigatus and A. flavus in solid organ transplant recipients. Mycopathologia. 2013; 176(5-6): 377-385.

3.            Line 41: “…. invasive pulmonary aspergillosis (IPA).” This sentence doesn’t have any related references. For suggestion, you can use this one too: b. Molecular Diversity of Aspergilli in Two Iranian Hospitals. Mycopathologia. 2021; 186:519–533.

4.            Lines 62-64: this sentence is unclear, and needs an English revision.

5.            Lines 76 and 79: what is the difference between BAL and BALF separately.

6.            Line 90: “High-risk patients” Could you please mention which groups?

7.            Line 110: “sample” to be “specimen” in all the text.

8.            Lines 111 and 112: these data should be mentioned in a Table too.

9.            Line 175: ASA and SOFA to be explained where the authors want to use them for the first time in the text.

10.        Lines 176-178: this sentence is unclear, and needs an English revision.

11.         Line 190: “Aspergillus fumigatus (A. fumigatus)” to be “A. fumigatus”.

12.        Lines 188-192: It is unclear by which methods these species are identified.

13.        Lines 193-194: this sentence is unclear, and needs an English revision.

14.        Lines 199-203: The authors should mention antifungal names in the related Tables too.

15.        Table 4 (contents) is unclear.

16.        Lines 256-259: there are no any Tables 5 and 6.

17.        Lines 260-262: this sentence is unclear, and needs an English revision.

18.        Line 275: “et al.” to be Italic in all the text.

19.        Line 313: “…. (e.g., GM) are necessary to overcome this diagnostic dilemma.” This sentence doesn’t have any related references. For suggestion, you can use this one too: c. Effect of involved Aspergillus species on galactomannan in bronchoalveolar lavage of patients with invasive aspergillosis. J Med Microbiol. 2017; 66: 898-904.

Author Response

Re: Manuscript ID: jcm-2276229

Impact of invasive pulmonary aspergillosis in critically ill surgical patients with or without solid organ transplantation

by

Simon Dubler, Michael Etringer, Markus A. Weigand, Thorsten Brenner, Stefan Zimmermann, Paul Schnitzler, Bettina Budeus, Fabian Rengier, Paulina Kalinowska, Yuan Lih Hoo and Christoph Lichtenstern

Response to the Reviewer`s suggestions                        Essen, 21st April 2023

Dear Editor,

Thank you very much for assessing our manuscript. We are very happy that our manuscript is potentially acceptable for publication in the journal “Journal of Clinical Medicine” after implementation of the latest Reviewer´s suggestions.

We thank both reviewers for their valuable comments, which really helped to improve the quality of our manuscript.

All issues mentioned by the Reviewer were considered very carefully and we would like to resubmit the revised version of the manuscript. You will find our responses to all issues and a detailed Point-by-Point analysis, explaining how the changes in the revised manuscript have been made.

If you have any further questions or suggestions, please do not hesitate to contact us.

We hope that we have satisfactorily addressed all the issues mentioned by the Reviewer and that the manuscript is now acceptable for publication in the journal “Journal of Clinical Medicine”.

Thank you very much for your helpful suggestions and for your kind support.

With kind regards from Essen

Simon Dubler

------------------------------------------------------------------------------------

Simon Dubler

Department of Anaesthesiology and intensive care medicine

Essen University Hospital

Hufelandstrasse 55

D-45147 Essen

Germany

Phone:                 +49-201 / 723-84053

Fax:                        +49 201 / 723-5949

Mail:                      Simon.Dubler@uk-essen.de

Reviewer #2

Detailed comments

Reviewer´s suggestion:

  1. Lines 28-32: this sentence is unclear, and needs an English revision.
  2. Line 41: “….highest risks of developing invasive aspergillosis (IA).” This sentence doesn’t have any related references. For suggestion, you can use this one too: a. Use of mycological, nested-PCR and real-time PCR methods on BAL fluids for detection of Aspergillus fumigatus and A. flavus in solid organ transplant recipients. Mycopathologia. 2013; 176(5-6): 377-385.
  3. Line 41: “…. invasive pulmonary aspergillosis (IPA).” This sentence doesn’t have any related references. For suggestion, you can use this one too: b. Molecular Diversity of Aspergilli in Two Iranian Hospitals. Mycopathologia. 2021; 186:519–533.
  4. Lines 62-64: this sentence is unclear, and needs an English revision.
  5. Lines 76 and 79: what is the difference between BAL and BALF separately.
  6. Line 90: “High-risk patients” Could you please mention which groups?
  7. Line 110: “sample” to be “specimen” in all the text.
  8. Lines 111 and 112: these data should be mentioned in a Table too.
  9. Line 175: ASA and SOFA to be explained where the authors want to use them for the first time in the text.
  10. Lines 176-178: this sentence is unclear, and needs an English revision.
  11. Line 190: “Aspergillus fumigatus (A. fumigatus)” to be “A. fumigatus”.
  12. Lines 188-192: It is unclear by which methods these species are identified.
  13. Lines 193-194: this sentence is unclear, and needs an English revision.
  14. Lines 199-203: The authors should mention antifungal names in the related Tables too.
  15. Table 4 (contents) is unclear.
  16. Lines 256-259: there are no any Tables 5 and 6.
  17. Lines 260-262: this sentence is unclear, and needs an English revision.
  18. Line 275: “et al.” to be Italic in all the text.
  19. Line 313: “…. (e.g., GM) are necessary to overcome this diagnostic dilemma.” This sentence doesn’t have any related references. For suggestion, you can use this one too: c. Effect of involved Aspergillus species on galactomannan in bronchoalveolar lavage of patients with invasive aspergillosis. J Med Microbiol. 2017; 66: 898-904.

Authors’ statement:

  1. We agree with the reviewer and we changed the sentence in the manuscript as follows.

  1. Revised text section, abstract, lines 28 - 30:

A higher Sequential Organ Failure Assessment (SOFA) score and evidence of bacteraemia were identified as risk factors for mortality and unfavourable outcome, whereas pIPA itself was not identified as an independent predictor for poor outcomes.

  1. We thank the reviewer for this comment and inserted the reference suggested by the reviewer.

Revised text section, page 1, line 42:

Additional Reference: (3) Zarrinfar H, Mirhendi H, Makimura K, Satoh K, Khodadadi H, Paknejad O. Use of mycological, nested PCR, and real-time PCR methods on BAL fluids for detection of Aspergillus fumigatus and A. flavus in solid organ transplant recipients. Mycopathologia. 2013;176(5-6):377-85.

  1. We agree with the reviewers ‘comment and added the suggested reference to the manuscript.

Revised text section, page 1, line 44:

Additional Reference: (4) Najafzadeh MJ, Dolatabadi S, Zarrinfar H, Houbraken J. Molecular Diversity of Aspergilli in Two Iranian Hospitals. Mycopathologia. 2021;186(4):519-33.

  1. We apologise for this. The sentence has been changed.

Revised text section, page 2, lines 59 - 62

An Italian multicentre observational study examined the presence of IFI in 18 ICUs over a period of 18 months. A total of 105 episodes of IFI occurred in 5,561 patients. Only 12 patients were diagnosed with invasive pulmonary aspergillosis, but mortality was high as 60%.

  1. We thank the reviewer for this comment. BAL (Broncho-alveolar lavage) is the procedure to collect BALF (broncho-alveolar lavage fluid). We think it is important to correctly address this procedure since procedures like tracheal aspirations could lead to false interpretations in case of Aspergillus spp. detection.

  1. We included an explanation for “high-risk patients”.

Revised text section, page 3, line 91 - 92:

High-risk patients [MELD (Model for Endstage Liver Disease) > 30; Re-liver transplantation and patients after high-urgency (HU) transplantation] received caspofungin for anti-fungal prophylaxis.

  1. We thank the reviewer for this important comment and changed “sample” to “specimen” in the manuscript.

  1. Revised text section, page 3, line 113: specimen
  2. Revised text section, page 9, line 385: specimen

  1. We thank the reviewer for this comment and included a figure with the diagnostic criteria of invasive pulmonary aspergillosis by Blot.

Revised text section, Supplementary materials.

  1. We thank the reviewer for this comment and explained SOFA and ASA in the main text of the manuscript.

Revised text section, page 5, line 169 – 170:

There were also differences in terms of the American Society of Anesthesiologists (ASA) status (p = 0.03) and Sequential Organ Failure Assessment (SOFA) scores at ICU admis-sion between pIPA- and non-pIPA patients (11 vs 8, p = 0.04).

  1. The mentioned sentence was revised.

Revised text section, page 5, line 171 - 173:

The length of ICU stay and the duration of mechanical ventilation tended to be prolonged in pIPA patients compared to non-pIPA patients [46 vs 30 days (p = 0.065) and 453 vs 270 hours, (p = 0.12) respectively].

  1. We apologize for this error and we corrected the mentioned word.

Revised text section, page 5, line 185:

A.fumigatus

  1. We thank the reviewer for this valuable comment. More details about the methods used were added in the Microbiology section of Methods.

Revised text sections, page 4, line 129 – 148:

GM testing was performed with the Platelia™ Aspergillus ELISA (Bio-Rad). Microbi-al growth in blood culture bottles (BACTEC FX® Aerobic/F (Ref. 442023), Lytic/10 Anaero-bic/F (Ref. 442021), BD Diagnostics, Sparks, United states) was detected by the BACTEC FX® automated blood culture system (BD Diagnostics) and subsequently confirmed by Gram staining. Positive cultures were worked up according to approved in-hospital standard techniques. Briefly, Columbia blood agar, chocolate agar and McConkey agar was used for plating of positive blood culture bottles, for Lytic/10 Anaerobic/F bottles an additional Schaedler/KV biplate (PB5204E, Thermofisher, Germany) was inoculated. If yeast or filamentous fungi were seen in the Gram stain, an additional CHROMagar™ Candida (254106, Becton Dickinson, Sparks, US) or Sabouraud Chloramphenicol™agar (254091, BD) was inoculated and incubated for 24-72 hours respectively.

Aspergillus spp. isolates were grown from respiratory specimens Isolates were inves-tigated at the Department of Medical Microbiology and Hygiene, bacteriology division of University Hospital Heidelberg, Heidelberg, Germany. Briefly, all respiratory specimens were inoculated on Sabouraud Chloramphenicol™agar (254091, BD) and incubated at 30°C degrees for 7 days with daily plate inspections. Only positive cultures with Aspergillus spp. were included in the study.

For yeast identification MALDI-TOF/MS identification was performed, as described previously (17). For filamentous fungi a PCR for the its region was performed followed by classical sequencing as described previously (18, 19).

  1. We apologize for this misunderstanding and corrected the sentence as following.

Revised text section, page 5, line 188 - 189

There was no difference in the use of antifungal prophylaxis between pIPA- and non-pIPA patients (23% vs 20%).

  1. We agree with the reviewers’ comment. We think it is only of value for the readers to add the antifungal to pIPA patients in the tables with non-SOT patients (Table S1a) and SOT patients (Table S1b). I hope the reviewer is satisfied with this.

Revised text section, supplementary materials (Table S1a and Table S1b)

  1. We apologize, that the reviewer did not understand the content of Table 4, which shows risk factors for death within 28 days after ICU admission in all included (SOT- and non-SOT) patients, where bacteremia was an independent risk factor for mortality (of all patients included in the study).

  1. We apologize for this inconvenience, but do not understand what the reviewer meant with Tables 5 and 6 concerning the sentence from Lines 256 – 259. Tables 5 and 6 are in suited in the supplementary material section. We preferred not to refer to tables during in the discussion. I hope the reviewer and the editor are happy with this.

  1. We thank the reviewer for this important comment and changed the word “blood stream infection (BSI)” for “bacteremia”.

  1. Revised text section, page 6, lines 265 - 266:

Of the 27 patients with bacteremia in our cohort, almost half of them (44% [12/27]) revealed an abdominal focus.

  1. Revised text section, page 6, lines 267:

This is not surprising since the cohort consisted of critically ill surgical patients at risk for bacteremia (19) due to postoperative surgical complications such as wound infections, anastomotic leakage, intra-abdominal abscesses or bowel ischemia with lethal complica-tions (20).

  1. Revised text section, page 7, lines 270:

This might have contributed to the high burden of bacteremia in our cohort.

  1. We agree with the reviewer and changed all “et al” in italic in the whole manuscript.

  1. Revised text section, page 7, Line 287
  2. Revised text section, page 7, Line 289
  3. Revised text section, page 7, Line 311

  1. The suggested reference was included in the manuscript.

Revised text section, page 7, line 286:

Reference 23: 23. Taghizadeh-Armaki M, Hedayati MT, Moqarabzadeh V, Ansari S, Mahdavi Omran S, Zarrinfar H, et al. Effect of involved Aspergillus species on galactomannan in bronchoalveolar lavage of patients with invasive aspergillosis. J Med Microbiol. 2017;66(7):898-904.
